# A Heuristic Algorithm for the Routing and Scheduling Problem with Time Windows: A Case Study of the Automotive Industry in Mexico

**Marco Antonio Juárez Pérez [1],\*** , **Rodolfo Eleazar Pérez Loaiza [1]** ,
**Perfecto Malaquias Quintero Flores [1]** , **Oscar Atriano Ponce [2]** and **Carolina Flores Peralta [1]**

[1]  Tecnológico Nacional de México, Instituto Tecnológico de Apizaco, Apizaco 90300, Mexico;
    rodolfo.pl@apizaco.tecnm.mx (R.E.P.L.); perfecto.qf@apizaco.tecnm.mx (P.M.Q.F.);
    carolinaflxp@gmail.com (C.F.P.)
[2]  Smartsoft America BA, Chiautempan 90802, Mexico; oatriano@smartsoftamerica.com.mx
\*  Correspondence: marcoantoniojuarezperez@gmail.com; Tel.: +52-241-149-6529

**Abstract:** This paper investigates a real-world distribution problem arising in the vehicle production industry, particularly in a logistics company, in which cars and vans must be loaded on auto-carriers and then delivered to dealerships. A solution to the problem involves the loading and optimal routing, without violating the capacity and time window constraints for each auto-carrier. A two-phase heuristic algorithm was implemented to solve the problem. In the first phase the heuristic builds a route with an optimal insertion procedure, and in the second phase the determination of a feasible loading. The experimental results show that the purposed algorithm can be used to tackle the transportation problem in terms of minimizing total traveling distance, loading/unloading operations and transportation costs, facilitating a decision-making process for the logistics company.

**Keywords:** heuristic; time windows; feasible loading; auto-carrier transportation problem (ACTP)

## 1. Introduction

Vehicle production in Mexico has been increasing in recent years [1,2]. As well as the number of imported vehicles, generating one of the main tasks to be solved by logistics companies: the transport of vehicles to dealerships. Currently, there are several commercial offers that provide a solution to route planning and fleet management. However, the cost of these applications is significantly high because they depend on the number of auto-carriers to route, making the acquisition of application difficult to afford.

This paper presents a vehicle-routing problem with time windows (VRPTW) in the real-world proposed by a logistics company in Mexico. In the problem, a heterogeneous fleet of auto-carriers departs from the new car storage yard (NCSY), delivers and unloads vehicles in the dealerships within predefined time windows, and finishes at the NCSY as shown in Figure 1. The objective of this research is to design and develop a logistic software to solve it. Considering the restrictions on the transport of vehicles imposed by Mexican traffic regulations [3], capacity, and allocation constraints, optimal performance with delivery time windows and proper planning of transportation routes. The software has a two-phase heuristic algorithm: in the first phase, the heuristic [4] is implemented to design the auto-carrier routes and an algorithm is proposed for the allocation of the vehicles in the auto-carriers. For experimentation, we used

a real database of approximately 4000 vehicles, more than 600 auto-carriers, and 44 different dealerships as a destination, obtaining results with the proposed algorithm in a reasonable time.

The structure of this paper is as follows. A review the relevant literature is provided in Section 2. An overview of the importance of research and the problem are shown in Section 3. The VRPTW is defined and mathematical formulations are presented in Section 4. The proposed solution and the developed methodology are described in Section 5. The experimental results of the algorithm are presented and analyzed with real-world instances in Section 6. Finally, in Section 7 conclusions and future research work are given.

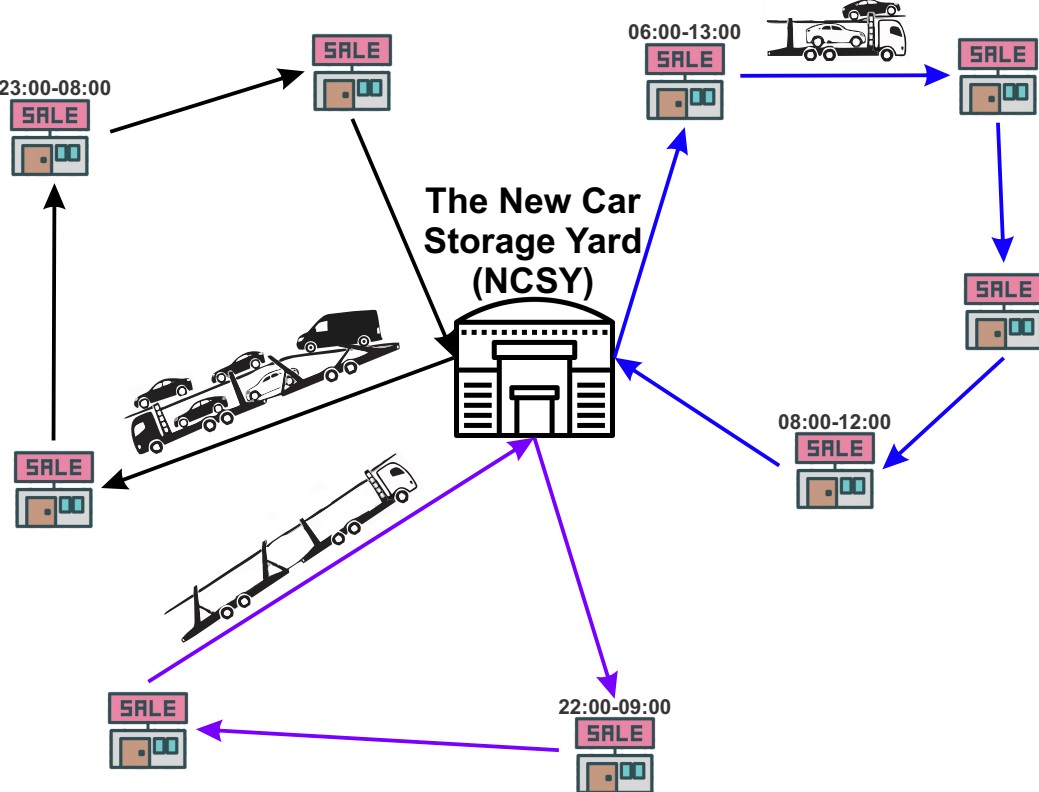

**Figure 1.** An illustration for the proposed problem by a logistics company.

## 2. Literature Review

The definition of the vehicle-routing problem (VRP) has its origins in the formulation of the traveling salesman problem (TSP) [5]. This section first reviews proposed algorithms and methods for the VRP and its variants. Then, focusing on the revision of the VRPTW and finally conclude with the review of the auto-carrier transportation problem (ACTP).

An important part of optimization systems are heuristics, which have multiple applications. From the extraction of features for a voice evaluation mechanism [6] to the generation of feasible VRP solutions. Arnau et al. [7] studied VRP with dynamic travel times, considering inputs of a dynamic nature and re-evaluating travel times dynamically as the solution was being developed. They proposed a learnt heuristic-based approach that integrates statistical learning techniques within a metaheuristic framework. Cassettari et al. [8] investigated the capacitated vehicle-routing problem (CVRP) applied to natural gas distribution networks. The authors introduced an algorithm based on the saving algorithm heuristic approach to solve it. Zhao and Lu [9] presented an electric vehicle-routing problem (EVRP) raised by a

logistics company. They developed a heuristic approach based on the adaptive large neighborhood search (ALNS) and integer programming, specifically designed a charging station heuristic adjustment and other one for the departure time decreasing the total operational cost.

Some well-known heuristic algorithms have been inspired by natural physical phenomena. Stodola [10] addressed the modified multi-depot vehicle-routing problem (MDVRP). He developed a metaheuristic algorithm based on the ant colony optimization (ACO) improved by a deterministic optimization process that is executed repeatedly within the ACO algorithm iterations. Połap and Woźniak [11] proposed a polar bear optimization algorithm (PBO) which imitates the survival and hunting behaviors of polar bears for local and global search. The authors presented a novel birth and death mechanism to control the population. Chen et al. [12] proposed a monarch butterfly optimization (MBO) algorithm to solve the dynamic vehicle-routing problem (DVRP) using a greedy strategy. Ahmed and Sun [13] designed a bilayer local search-based particle swarm optimization (BLS-PSO) algorithm to solve CVRP.

Currently, one of the most studied variants of the VRP is with time windows, in the research by Desrochers et al. [14] introduced an optimization algorithm to solve a VRPTW, using dynamic programming. Tan et al. [15] explored simulated annealing (SA), tabu search (TS) and a genetic algorithm (GA) heuristics to solve it. In another study, Yu et al. [16] proposed a hybrid approach, consisting of the use of the ACO and TS algorithms, for the VRPTW. To improve the performance of the ACO algorithm, they introduced a neighborhood search and a TS algorithm to maintain the diversity of the ACO algorithm and explore new solutions. Taner et al. [17] developed two metaheuristic algorithms to solve the VRPTW, the SA algorithm and an iterated local search (ILS). Sripriya et al. [18] designed a hybrid genetic search with diversity control using a GA to solve the VRPTW, using the Pareto approach and two mutation operators to find the optimal solution set.

Tadei et al. [19] investigated and defined a variant of the VRP, called the ACTP, proposed a three-step heuristic procedure that considers the loading, vehicle selection, and routing aspects for a solution to the problem. In other research, B. M. Miller [20] addressed the ACTP for collection and delivery with limitations in the delivery times and the capacity of the auto-carrier, for new and used vehicles. The author proposed a constructive heuristic to solve the problem. Dell'Amico, et al. [21] defined the ACTP as a combinatorial problem of the CVRP. The authors presented a study of a real case and implemented an ILS algorithm for the routing and mathematical techniques for the loading of vehicles. On the other hand, Tran et al. [22] implemented a heuristic algorithm for location of alternative-fuel stations. Hosseinabadi et al. [23] developed a method called TIME_GELS that uses the gravitational emulation local search algorithm (GELS) for solving the multiobjective flexible dynamic job-shop scheduling problem.

The VRP is widely studied in the areas of operations research and computer sciences, due to its computational complexity and its multiple applications. The variants of the VRP allow the use of time window restrictions and vehicle capacity, among others, these restrictions allow solving problems with solutions closer to the optimum of real-world cases, the ACTP is a result from this. As it has been described in the literature, several authors have proposed algorithms and methods to solve this problem. Nevertheless, the characteristics of our problem, motivate us to implement a heuristic approach that contemplates the restrictions imposed by the logistics company.

## 3. Importance of the Problem

The automotive industry has been one of the most important engines for the development and economic growth of Mexico [2,24]. Hence, the importance of promoting the insertion of technology in the sectors that are parts of it. For this reason, it is necessary to implement technology in the process of

transporting new vehicles within the country. In addition, with it to diminish one of the most expensive processes for the companies of transport, the routing and scheduling of auto-carriers.

In addition to having a positive impact on the operating expenses for the transportation companies, decreasing the amount kilometers of the traveled route from each of the auto-carrier also represents a positive environmental impact because downward the harmful emissions to the environment produced by diesel motors. According to [25] most of the auto-carriers use this fuel and the main characteristic of diesel emissions is that particles are produced in a proportion 20 times higher than gasoline engines.

Nitrogen oxides ($NO_x$) are considered an important source of air pollution and contribute greatly to photochemical smog, acid rain, depletion of the ozone layer and the greenhouse effect. Diesel exhaust gases are generally composed of more than 90% $NO_x$ [26]. Therefore, one of the main contributors to emissions of $NO_x$ and sulfur oxides are diesel engines [27], so it is important to reduce these emissions. Optimizing the routes of the auto-carriers that generate these emissions are a good way to do it.

According to the National Institute of Statistics and Geography [24], the Mexican automotive industry is important because:

- It is ranked as the second most important activity in manufacturing after the food industry
- Because its exports were ranked fourth in the world in 2014
- When demanding inputs to carry out its production, it generates impacts on 157 economic activities out of a total of 259, according to the input-output matrix

The production of the automotive industry has increased its relative importance in the economy. Before the North American Free Trade Agreement (NAFTA) came into force, this industry represented 1.9% of gross domestic product (GDP) in Mexico, while in 2014 it was 3.0% [24]. This increase was due to the implementation of new technologies in the last decade. Both in the automotive sector and in the rest of economic activities that are suppliers of this, one of them the transportation of vehicles.

Finally, the transportation of vehicles is an important field in operations research (OR), which has attracted increasing interest in recent years, due to the expected benefits of substantial cost reduction and efficient consumption of resources. The VRPTW has multiple applications such as supermarkets, cement plants, hospitals, etc., though its main applications are in the industry.

## 4. Problem Definition and Mathematical Model

A logistics company distributes new vehicles in Mexico, manufactured in another country. It carries out the delivery of thousands of vehicles, according to the demand of each of the dealerships responsible for the sale of vehicles. Currently, the logistics company uses an empirical allocation and routing method for the auto-carriers.

The empirical method consists of the design of the route according to the experience of the operator of the auto-carrier, based on the vehicles that will be transported without the use of a heuristic or similar method for the optimization of the route. Similarly, the allocation corresponds to a simulation with fictitious vehicles of the load of the auto-carrier, positioning the vehicles in different levels of the auto-carrier, considering the dimensional restrictions.

The process of the routing and loading of the auto-carriers, begins at the moment that the operators receive a list of the vehicles to be delivered to the different dealerships. In the NCSY, the operators confirm the vehicles to be transported with the manager of the NCSY, who to complete the loading process verifies that the vehicles in the auto-carrier correspond to the request of the dealerships.

Empirical routing is inefficient because it does not consider restrictions as the time windows. The time windows are the hours in which an auto-carrier can perform the unloading of vehicles at a dealership, who defines an initial time and an end time to carry it out. Time windows are defined to not violate local traffic laws, thus avoiding monetary penalties.

Avoiding various penalties and monetary losses for companies, are some reasons of importance for the VRPTW and its applications. An example of its application is in the cement industry, if the concrete mixer trucks do not arrive within the stipulated time window. It may be that part of the concrete dries, becoming unusable, and the work stops. In the case of the logistics company, if an auto-carrier arrives at the dealership at a time outside the time window, it causes a time penalty that is, the operator must wait at the dealership to unload vehicles.

In addition to the time window restrictions, this case study includes a total of 44 dealerships, a demand with approximately 4000 vehicles of different dimensions (which add three restrictions to the allocation) and a variable number of auto-carriers of different capacity load (3, 6, 7, 10 and 11 vehicles). This paper describes the algorithm developed for a real-world problem of a logistics company; the problem can be summarized as follows:

> given a heterogeneous fleet of auto-carriers based at a NCSY and a set of dealerships each requiring a set of vehicles, the loading of the vehicles into the auto-carriers and route the auto-carriers through the road network to deliver all dealerships with minimum cost (total number of kilometers traveled) that start and ends in the NCSY, considering the restrictions of time windows, a LIFO policy for the loading/unloading of vehicles and maximizing the total use of the capacity of each auto-carrier

The characteristics of the dealerships (time windows), the NCSY and the auto-carriers (capacity), as well as different operational restrictions on the routes, bring forth the VRPTW, several authors [14–18] have worked on this variant of the VRP. In this case, study the term vehicle denotes a transported item (e.g., a car, a van), the term auto-carrier denotes a truck transporting vehicles, and the term dealership denotes a delivery point (i.e., a customer requiring one or more vehicles). With the previously mentioned elements, the model can be described as follows [21]:

- Network: Given a complete graph $G = (V, E)$, where $V = 0, 1, \ldots, n$ is the set of vertices and $E$ the set of edges connecting each vertex pair. Vertex 0 corresponds to the NCSY, whereas vertices $1, \ldots, n$ correspond to the $n$ dealerships to be served. The edge is connecting vertices $i$ and $j$ is denoted by $(i, j)$ and has an associated routing cost $c_{ij}(i, j \in V)$ shown in Figure 2. The distance and times matrices are symmetric.

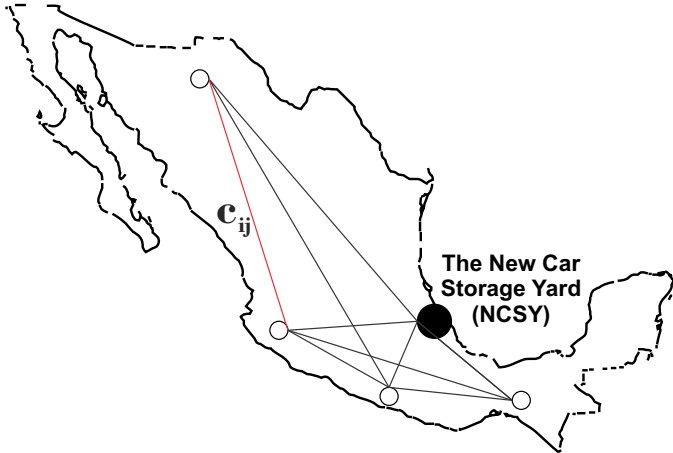

**Figure 2.** Example for routing cost between dealership $i$ and $j$.

- Fleet: Given a heterogeneous fleet of auto-carriers, composed by a set $T$ of auto-carrier types. Each auto-carrier type $t(t \in T)$ has a maximum vehicles capacity $W_t$ and is formed $A_k^{1,2}$ by loading platforms (levels, shown in Figure 3). There are $K_t$ auto-carriers available for each type $t$.

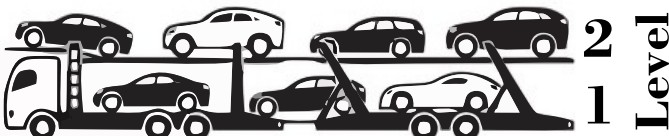

**Figure 3.** Auto-carrier levels.

- Demand: The demand of dealership $i$ consists of a set $M$ of vehicles ($i \in V \backslash \{0\}$). Each vehicle $m_i \in M$ demanded by dealership $i$ belongs to a vehicle type (or vehicle model) shown in Figure 4, which is defined by a height $h_m$ and a vehicle identification number (VIN).

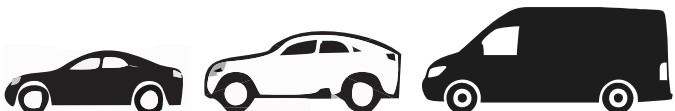

**Figure 4.** Vehicle types (vehicle, vehicle, van).

In this VRPTW, each dealership $i \in V \backslash \{0\}$ has an associated time window $[e_i, l_i]$, with a time allowed service for arriving auto-carriers to it and service time or delay $d_i$. If $(i, j)$ is an arc of the solution and $a_i$ and $a_j$ are the arrival times to the dealerships $i$ and $j$, time window imply that necessarily must be fulfilled $a_i \leq l_i$ and $a_j \leq l_j$. On the other hand, if $a_i \leq e_i$, then the auto-carrier must wait until the dealership "opens" so necessarily $a_j = e_i + d_i + c_{ij}$.

Using the nodes $0$ and $n + 1$ to represent the NCSY and the set $K$ to represent the auto-carriers, the problem is formulated for a heterogeneous fleet of auto-carriers, according to [28]:

$$\min \sum_{k \in K} \sum_{(i,j) \in E} c_{ij}^k x_{ij}^k \tag{1}$$

subject to

$$\sum_{k \in K} \sum_{j \in \Delta-(i)} x_{ij}^k = 1 \qquad \forall i \in V \backslash \{0, n+1\} \tag{2}$$

$$\sum_{j \in \Delta+(0)} x_{0j}^k = 1 \qquad \forall k \in K \tag{3}$$

$$\sum_{j \in \Delta+(i)} x_{ij}^k - \sum_{j \in \Delta-(i)} x_{ji}^k = 0 \qquad \forall k \in K, i \in V \backslash \{0, n+1\} \tag{4}$$

$$\sum_{i \in V \backslash \{0, n+1\}} x_{ij}^k - \sum_{j \in \Delta+(i)} x_{ji}^k \leq W^k \qquad \forall k \in K \tag{5}$$

$$y_j^k - y_i^k \geq d_i + a_{ij}^k - H(1 - x_{ij}^k) \qquad \forall i, j \in V \backslash \{0, n+1\}, k \in K \tag{6}$$

$$e_i \leq y_i^k \leq l_i \qquad \forall i \in V \backslash \{0, n+1\}, k \in K \tag{7}$$

$$x_{ij}^k \in 0, 1 \qquad \forall (i, j) \in E, k \in K$$

$$y_i^k \geq 0 \qquad \forall i \in V \setminus \{0, n+1\}, k \in K$$

The $x_{ij}^k$ variables indicate if the arc $(i, j)$ is used by the auto-carrier $k$. The $y_i^k$ variables indicate the arrival time at the dealership $i$ when it is visited by the $k$ auto-carrier (if the dealership is not visited by the auto-carrier, the variable has no meaning). The objective function (1) minimizes the total routing cost.

Constraint (2) state that each dealership is visited exactly once, while constraints (3) and (4) determine that each auto-carrier $k \in K$ goes through a path of 0 to $n+1$. The capacity of each auto-carrier is imposed in (5). Since $H$ is a sufficiently large constant, restriction (6) ensures that if an $k$ auto-carrier travels from $i$ to $j$, it cannot reach $j$ before $y_i + d_i + a_{ij}^k$. These constraints also eliminate subtours and constraints (7) enforce time windows restriction.

The use of a heterogeneous fleet and the nature of the demand (vehicles and vans) impose the allocation constraints:

$$h_m > 2.5 = (A_k^{1-2}, W_k^3) \qquad \forall m_i \in M, k \in K \tag{8}$$

$$1.8 < h_m < 2.5 = (A_k^1, W_k^1) \qquad \forall m_i \in M, k \in K \tag{9}$$

$$h_m < 1.8 = (A_k^{1,2}, W_k^1) \qquad \forall m_i \in M, k \in K \tag{10}$$

The constraint (8) considers the assignment of a vehicle $m$ with a height $h_m$ greater than 2.5 m (meters) that corresponds to a van, which occupies an allocated space in level $A_k^1$ and two spaces on level $A_k^2$, using three spaces of the capacity $W$ of the auto-carrier $k$. A vehicle $m$ with height $h_m$ greater than 1.8 m and less than 2.5 m, its assignment corresponds to a space $W_k^1$ and can only be accommodated at level $A_k^1$, i.e., the constraint (9). The last assignment constraint (10) defines that a vehicle $m$ with a height $h_m$ less than 1.8 m corresponds to an allocation space $W_k^1$ and can be accommodated in either of the two levels $A_k^1$ or $A_k^2$.

## 5. Methodology

The use of a heuristic methodology allows obtaining the solution to the routing problem of the auto-carriers at a reasonable time, meaning a representative change versus the empirical methodology previously used by the logistics company. A graphic illustration of the comparison of the insertion heuristic I1 is made in [29,30]. Regarding its comparison with other methods is presented in [31] and its computational complexity of the proposed algorithm is $O(n^2 \log n^2)$. Hereafter, the approach and development of the heuristic algorithm are described.

### 5.1. Heuristic Approach

The development of the solution is divided into two phases, the first one is to generate the route of the auto-carrier and the second one the vehicle allocation in the auto-carrier, both phases are part of a main algorithm. In the first phase, the routes are obtained with the implementation of the Solomon I1 insertion heuristic, due to the logistics company is needed to obtain a solution to the VRPTW in fairly necessary time, given that the VRPTW is an NP-complete problem [28]. This routing process applies a methodology of cluster first, route second, i.e., first group by the dealership, to then build the route, which starts with the dealership that has the shortest and earliest time window, considering the allocation and capacity constraints. The following describes the application of the Solomon I1 insertion heuristic [4] and the vehicle loading process for allocation phase on this VRPTW.

The routing algorithm builds a feasible solution by constructing one route at a time. At each iteration the algorithm decides which new dealership $u^* \in U$ has to be inserted in the current solution, and between which adjacent dealerships $i(u^*)$ and $j(u^*)$ the new dealership $u^*$ has to be inserted on the current route. When choosing $u^*$, the algorithm takes into account both the cost increase associated with the insertion of $u^*$, and the delay in service time at dealerships following $u^*$ on the route. The three steps of the algorithm are:

**Step 0**. *(Initialization)*. The first route is initially $R_1 = \{0, i, 0\}$, where $i$ is the dealership with the shortest and earliest time window. In the allocation phase, if vehicle $m$ of dealership $i$ has a feasible assignment, then set $k = 1$, otherwise get the next vehicle from dealership $i$ or next dealership with the shortest and earliest time window, until the allocation phase of the vehicle $m$ is feasible.

**Step 1**. Let $R_k = \{i_0, i_1, \ldots, i_m\}$ be the current route, where $i_0 = i_m = 0$, i.e., the NCSY. Set

$$f1(i_{p-1}, u, i_p) = \alpha(r_{i_{p-1}u} + r_{ui_p} - \mu r_{i_{p-1}i_p}) + (1 - \alpha)(b^u_{i_p} - b_{i_p}) \tag{11}$$

where $0 \leq \alpha \leq 1$, $\mu \geq 0$ and $b^u_{i_p}$ is the time when service begins at dealership $i_p$ provided that dealership $u$ is inserted between $i_{p-1}$ and $i_p$. For each unrouted dealership $u$, compute its best feasible insertion position in route $R_k$ as:

$$f1(i(u), u, j(u)) = \min_{p=1,\ldots,m} f1(i_{p-1}, u, i_p)$$

where $i(u)$ and $j(u)$ are the two adjacent vertices of the current route between which $u$ should be inserted. Determine the best unrouted customer $u^*$ to be inserted yielding.

$$f2(i(u^*), u^*, j(u^*)) = \max_u \{f2(i(u), u, j(u))\}$$

where

$$f2(i(u), u, j(u)) = \lambda r_{0_u} - f1(i(u), u, j(u)) \tag{12}$$

with $\lambda \geq 0$.

**Step 2**. Insert dealership $u^*$ in route $R_k$ between $i(u^*)$ and $j(u^*)$, in the allocation phase, if vehicle $m$ of dealership $u^*$ has a feasible assignment, then go back to **Step 1**, otherwise get the next vehicle from dealership $u^*$ until the allocation phase of the vehicle $m$ is feasible. If $u^*$ does not exist, but there are still unrouted dealerships, set $k = k + 1$, initialize a new route $R_k$ (as in **Step 0**) and go back to **Step 1**. Otherwise, **STOP**, a feasible solution has been found.

The insertion heuristic tries to maximize the benefit obtained when servicing a dealership on the current route rather than on an individual route. For example, when $\mu = \alpha = \lambda = 1$, Equation (12) corresponds to the saving in distance from servicing dealership $u$ on the same route as dealerships $i$ and $j$ rather than using an individual route. The best feasible insertion place of an unrouted dealership is determined by minimizing a measure, defined by the Equation (11), of the extra distance and the extra time required to visit it. Different values of the parameters $\mu$, $\alpha$ and $\lambda$ lead to different possible criteria for selecting the dealership to be inserted and its best position in the current route.

After starting a new route $R_k$ or inserting a dealership $u^*$ in the current route, the vehicle allocation phase is responsible for obtaining the feasible load of the auto-carrier, considering the constraints imposed by the logistics company, which are listed below by rank:

- Vehicle $h_m > 2.5$ m: It uses three spaces of the capacity of the auto-carrier $k$, i.e., a space in level $A^1_k$ and two spaces on level $A^2_k$, this is shown in Figure 5. To maximize the use of the capacity of the auto-carrier, another allocation is to occupy one space above ($A^2_k$) and two below ($A^1_k$).

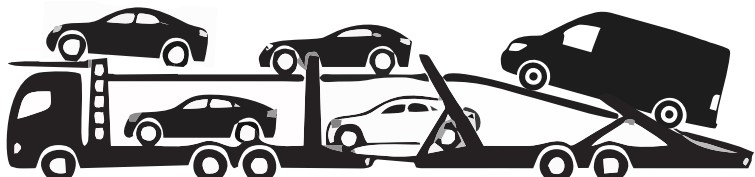

**Figure 5.** Constraint of vehicles with $h > 2.5$ m.

- Vehicle 1.8 m $< h_m < 2.5$ m: It uses a space of the capacity of the auto-carrier $k$ and can only be assigned in level $A_k^1$, as shown in Figure 6.

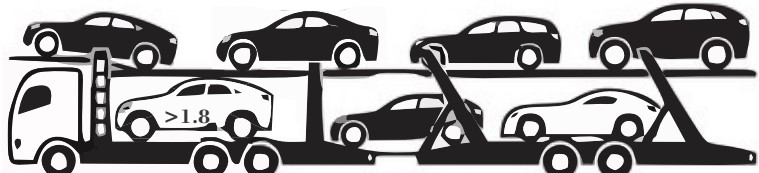

**Figure 6.** Constraint of vehicles with 1.8m $< h < 2.5$ m.

- Vehicle $h_m < 1.8$ m: It uses a space of the capacity of the auto-carrier $k$ and can be assigned in any available space to it, as shown in Figure 7.

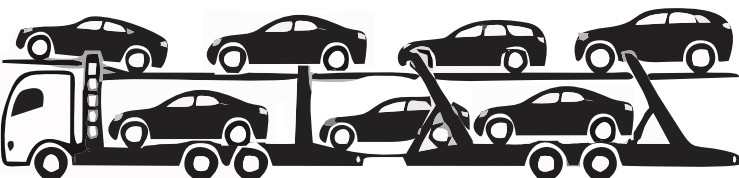

**Figure 7.** Constraint of vehicles with $h < 1.8$ m.

- Policy Last In First Out (LIFO): Last vehicle loaded, first vehicle unloaded. For example, if the first dealership to visit is $d_2$ on the current route, the vehicles of $d_2$ should be the last to be loaded on the auto-carrier.

If the assignment of a vehicle $m$ is not feasible and there are still vehicles on demand, then go back to the routing phase, while the vehicle $m$ will be assigned to the next auto-carrier route. The development of the heuristic algorithm is described in the following subsection.

*5.2. Development of the Two-Phase Heuristic*

To implement the heuristic algorithm, it was necessary to create a distance matrix, with the distance information (in kilometers) among the 44 dealerships, as shown in Table 1, the NCSY is represented by $d_0$, e.g., a trip from the NCSY ($d_0$) to dealership 2 ($d_2$) has a cost of 1373 km, while the trip from dealership 2 ($d_2$) to dealership 44 ($d_{44}$) would represent a route of 1245 km.

**Table 1.** Distance (in kilometers) between dealerships.

| Dealership | $d_0$ | $d_1$ | $d_2$ | $d_{...}$ | $d_{44}$ |
|---|---|---|---|---|---|
| $d_0$ | 0 | 885 | 1373 | ... | 114 |
| $d_1$ | 885 | 0 | 498 | ... | 752 |
| $d_2$ | 1373 | 498 | 0 | ... | 1245 |
| $d_{\vdots}$ | ⋮ | ⋮ | ⋮ | 0 | ⋮ |
| $d_{44}$ | 114 | 752 | 1245 | ... | 0 |

A time matrix is also required for the implementation of the heuristic algorithm. Table 2 shows the duration in minutes of the travel times between the dealerships, for example, the duration of the trip from NCSY ($d_0$) to dealership 1 ($d_1$) is 568 min, in other words, 9 h and 28 min. A route from the dealership 44 ($d_{44}$) to dealership 2 ($d_2$) is 13 h and 49 min of travel.

**Table 2.** Travel times (in minutes) between dealerships.

| Dealership | $d_0$ | $d_1$ | $d_2$ | $d_{...}$ | $d_{44}$ |
|---|---|---|---|---|---|
| $d_0$ | 0 | 568 | 880 | ... | 92 |
| $d_1$ | 568 | 0 | 319 | ... | 490 |
| $d_2$ | 880 | 319 | 0 | ... | 829 |
| $d_{\vdots}$ | ⋮ | ⋮ | ⋮ | 0 | ⋮ |
| $d_{44}$ | 92 | 490 | 829 | ... | 0 |

The distances (Table 1) and times (Table 2) matrices are symmetric, but in the time windows, a matrix was created with the earliest ($e_i$) and the latest ($l_i$) time window, using a 24-h time format, as shown in Table 3. The NCSY ($d_0$) does not have a time window established, therefore, $e_i$ = 00:00 and $l_i$ = 23:59. Figure 1 shows an example of the dealerships who have established a time window, otherwise they do not have a time window established as the NCSY.

**Table 3.** Time windows for the dealerships in 24-h time format.

| | *TimeWindow* | |
|---|---|---|
| *Dealership* | $e_i$ | $l_i$ |
| $d_0$ | 00:00 | 23:59 |
| $d_1$ | 06:00 | 13:00 |
| $d_2$ | 08:00 | 12:00 |
| $d_{\vdots}$ | ⋮ | ⋮ |
| $d_{44}$ | 22:00 | 09:00 |

The heuristic is described in Algorithm 1, first phase is responsible for generating the route for the auto-carriers and the second of the feasible load of vehicles in the auto-carrier. This algorithm is codified in JAVA language, it has as input a matrix $M$ with the demand of vehicles to be transported and a matrix with auto-carriers $K$ available for the delivery of vehicles. The first phase clusters the demand $M$ according to the dealerships to visit ($U$), then perform the sorting of the dealerships with the shortest and earliest time window, considering that the execution time (*current time, CT*) of the algorithm influences this ordination,

i.e., the execution of the algorithm at different times of the day with the same data produces different routing outputs and vehicle accommodation.

---

**Algorithm 1:** Two-phase heuristic

---

   **Data:** $M$ (*demand*), $K$ (auto-carriers).

   **Output:** The $K$ auto-carriers with $A_k$ arrangement, $R_k$ route and delivery schedules. A vector with
       the remaining $M$, if any.

1  **begin**

2     $U \longleftarrow$ get *dealerships* from demand

3     $CT \longleftarrow$ get *current time*

4     **while** $M > 0$ **and** $K > 0$ **do**

5        Initialization a new route $R_k$

6        $W_t \longleftarrow$ get *capacity* from $k$

7        $A_k \longleftarrow$ generate a new arrangement with capacity $W_t$

8        $u \longleftarrow$ get *first dealership* to visit from $U$

9        **while** $W_t > 0$ **and** $u^* \in U$ **do**

10          **while** $m_u \in M$ **do**

11            **if** $Allocation(m_u, A_k)$ **then**

12              Update demand $M$, capacity $W_t$ and route $R_k$

13            **end**

14          **end**

15          $u^* \longleftarrow$ get *next dealership*$(R_k, CT)$ to visit and update $U$

16        **end**

17        **if** $K \le 0$ **and** $M > 0$ **then**

18          Get *remaining* from $M$

19        **end**

20        Add to *Solution*$(R_k, A_k)$, $k = k + 1$

21     **end**

22  **end**

---

The first loop is the demand $M$ and the auto-carriers $K$, while there are vehicles to load and auto-carriers, a new route $R_k$ is initialized with time and distance counters, the auto-carrier $k \in K$ is obtained, assigns $W_t$ according to $k$, which is the vehicle load capacity of $k$ for its type $t$, then a new accommodation $A_k$ based on capacity $W_t$ is generated. After determining the *first dealership* to visit $u$, i.e., Step 0 of the heuristic, this is shown in line 8 of Algorithm 1.

With the first dealership $u$ to be selected, start the loop of the auto-carrier $k$ with capacity $W_t$ and loop of the existing demand $M$ of the dealership $u$, carrying out the loading of the vehicle $m_u$ in the auto-carrier $k$ in the second phase of the algorithm, *the allocation*, this is shown in line 11 of Algorithm 1. The allocation algorithm receives as parameters the vector $A_k$ of the current arrangement and the vehicle $m_u$ to be loaded (see Algorithm 2). If the load is successful, then update the demand $M$ (eliminating $m_u$), the number of available spaces of the capacity $W_t$ and the route $R_k$. In the case that the auto-carrier $k$ is not filled or the vehicle $m_u$ does not comply with the assignment restrictions, obtain the next vehicle $m_u + 1$ to load, until the auto-carrier $k$ is full.

---

**Algorithm 2:** Allocation

---

**Data:** $m_u$ (vehicle), $A_k$ (arrangement).

**Output:** The $A_k$ arrangement with $m_u$ assigned if it is feasible.

1 **begin**

2      $W_t \longleftarrow$ size of $A_k$

3      $level \longleftarrow \frac{W_t}{2}$

4      **for** $A_{k_i} \in A_k$ **do**

5          **if** $h_{m_u} > 2.5$ **and** $i < level$ **and** *parity* **then**

6              **if** $A_{k_i} \in A_k$ **and** $A_{k_{i+1}} \in A_k$ **and** $A_{k_{i+level}} \in A_k$ **then**

7                  Allocate $m_u$ vehicle to spaces $A_{k_i}$, $A_{k_{i+1}}$ and $A_{k_{i+level}}$

8              **else**

9                  **if** $A_{k_i} \in A_k$ **and** $A_{k_{i+(level-1)}} \in A_k$ **and** $A_{k_{i+level}} \in A_k$ **then**

10                      Allocate $m_u$ vehicle to spaces $A_{k_i}$, $A_{k_{i+(level-1)}}$ and $A_{k_{i+level}}$

11                  **end**

12              **end**

13          **else**

14              **if** $A_{k_i} \in A_k$ **and** $A_{k_{i+1}} \in A_k$ **and** $A_{k_{i+(level+1)}} \in A_k$ **then**

15                  Allocate $m_u$ vehicle to spaces $A_{k_i}$, $A_{k_{i+1}}$ and $A_{k_{i+(level+1)}}$

16              **else**

17                  **if** $A_{k_i} \in A_k$ **and** $A_{k_{i+level}} \in A_k$ **and** $A_{k_{i+(level+1)}} \in A_k$ **then**

18                      Allocate $m_u$ vehicle to spaces $A_{k_i}$, $A_{k_{i+level}}$ and $A_{k_{i+(level+1)}}$

19                  **end**

20              **end**

21          **end**

22          **if** $h_{m_u} > 1.8$ **and** $h_{m_u} < 2.5$ **and** $i > level$ **and** $A_{k_i} \in A_k$ **then**

23              Allocate $m_u$ vehicle in space $A_{k_i}$

24          **end**

25          **if** $h_{m_u} < 1.8$ **and** $A_{k_i} \in A_k$ **then**

26              Allocate $m_u$ vehicle in space $A_{k_i}$

27          **end**

28      **end**

29      **return** $A_k$

30 **end**

---

If the auto-carrier $k$ still has available spaces $W_t$, but the demand of the dealership $u$ does not comply with the assignment restrictions, update accumulators of time and distance to obtain the next dealership $u^*$ to visit, this is Step 1 of the heuristic and is observed on line 15 of Algorithm 1. To obtain $u^*$ the route $R_k$ built so far and the current time $CT$ are received as parameters, finally enter the demand loop $M$ of the dealership $u^*$.

Once selected $u^*$, in the capacity loop $W_t$ the vehicle $m_{u^*}$ allocation phase of the auto-carrier $k$ starts, this is Step 2 of the heuristic algorithm. If the accommodation of $m_{u^*}$ is feasible and there are still spaces of $W_t$, return to Step 1 of the heuristic algorithm, to obtain the next vehicle $m_{u^*} + 1$, the capacity loop ends when $W_t$ is equal to 0 or does not exist $u^*$. Then, the route $R_k$ and the arrangement $A_k$ of the auto-carrier $k$ add to the *Solution*, and finally a new route $R_{k+1}$ is started.

A requirement of the logistics company is to add to the solution the remaining demand $M$, in the case that the number of auto-carriers $K$ were not enough to transport the demand $M$, line 17. Algorithm 1 ends when there is no demand $M$ or auto-carriers $K$ available for routing.

In the allocation phase (Algorithm 2), to accommodate the vehicles in the auto-carrier, $A_k$ is abstracted as a vector of size $W_t$ (capacity of the $t$-type auto-carrier), to simulate and delimit the levels of the auto-carrier a variable called *level* is created. If $t$ is pair, the index $i$ of the upper level initializes at $i = 0$ and ends at $i = level - 1$, while the lower level initializes at $i = level$ and ends at $i = W_t$, as shown in Figure 8a. If $t$ is odd, the index $i$ of the upper level initializes at $i = 0$ and ends at $i = level$, while the lower level initializes at $i = level + 1$ and ends at $i = W_t$, as shown in Figure 8b.

Once $A_k$ is defined, the load and assignment of the vehicle $m_u$ is defined by its height $h_{m_u}$, Algorithm 2 starts by obtaining $level = \frac{W_t}{2}$. If $h_{m_u} > 2.5$ m, the vehicle $m_u$ will occupy spaces in the two levels of the auto-carrier, then first determine if there is available space in the lower level $i < level$, otherwise $m_u$ is assigned in the next auto-carrier with available space, the case that is met $i < level$, *parity* verifies if $t$ is even or odd, depending on this result obtain the spaces that comply with Equation (8) and if are available, perform the allocation of $m_u$ (e.g., see Figure 5) to these spaces.

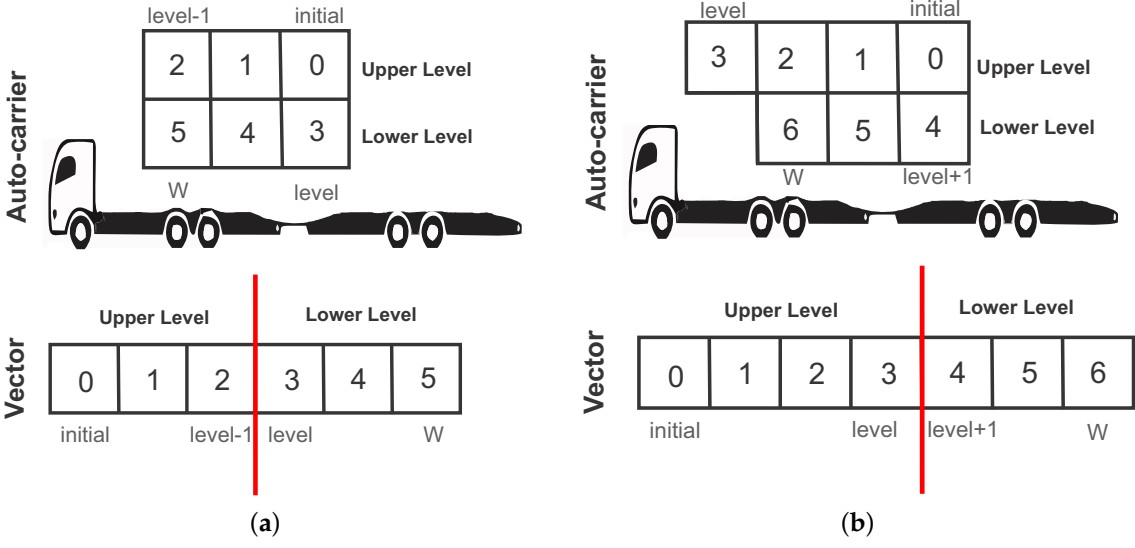

**Figure 8.** Vector of auto-carrier capacity: (**a**) Pair (**b**) Odd.

If $h_{m_u} > 1.8$ m and $h_{m_u} < 2.5$ m, to assign the vehicle $m_u$ verify if there is available space in the lower level $i > level$ as shown in line 22 of Algorithm 2, otherwise $m_u$ is assigned to the next auto-carrier with available space in the lower level. Finally, for $h_{m_u} < 1.8$ m it is only determined if there is space available in the auto-carrier and $m_u$ is allocated, with the assignment of $m_u$ in any of the cases and the return of $A_k$, the allocation phase ends.

## 6. Results and Analysis

### 6.1. Experimental Results

To evaluate the performance of the auto-carriers routing algorithm with time windows, two scenarios were designed with 11 different instances of the demand, the last instance corresponds to the real problem of the logistics company. The scenarios are the following:

- *Random Dealerships with Time Windows (RDTW)*—context in which most of the dealerships (34 of 44) were set different time windows for vehicle unloading.
- *Main Dealerships with Time Windows (MDTW)*—context that corresponds to the case of the logistics company, only the dealerships (14 of 44) that are located in the main cities of the country establish a time window for the unloading of vehicles.

The configuration of the most important parameters for the implementation of the proposed heuristic algorithm is shown in Table 4. Next, the content of the instances is described in Table 5, the first column corresponds to the instance number, the second is the demand size, and in the following columns the content of this in terms of vehicles (cars, partners) and vans (managers). It is necessary to emphasize the number of vans because they use more spaces in the auto-carrier compared to the vehicles. For both scenarios, the same instances demand was used to perform tests and compare the results of the total distance of the generated routes.

**Table 4.** Parameters of the proposed heuristic algorithm.

| Parameter | Value |
|---|---|
| **Algorithm 1** | |
| mu | 1 |
| alpha | 0.9 |
| lamda | 1 |
| time_unloading | 15 |
| k | 1 |
| **Algorithm 2** | |
| initial | 0 |
| $W_t$ | It depends on the auto-carrier K |
| level | $\frac{W_t}{2}$ |
| parity | It depends on the auto-carrier K |

**Table 5.** Test Instances.

| Instance | Demand Size | Cars | Partners | Managers |
|---|---|---|---|---|
| 1 | 20 | 6 | 14 | 0 |
| 2 | 50 | 46 | 1 | 3 |
| 3 | 100 | 32 | 66 | 2 |
| 4 | 200 | 108 | 87 | 5 |
| 5 | 500 | 206 | 270 | 24 |
| 6 | 1000 | 456 | 502 | 42 |
| 7 | 1500 | 654 | 767 | 79 |
| 8 | 2000 | 941 | 974 | 85 |
| 9 | 2500 | 1164 | 1224 | 112 |
| 10 | 3000 | 1433 | 1431 | 136 |
| 11 | 3884 | 1810 | 1906 | 168 |

With the instances of Table 5, a total of 132 tests were made in the two scenarios to the Algorithm 1, as a result of each of the tests the routes were obtained (auto-carriers, each route corresponds to one scenario previously mentioned) and the accommodation of the auto-carrier considering the allocation restrictions, in order to present all the results, these are grouped according to the capacity of the auto-carriers (3, 6, 7, 10, 11, and a heterogeneous fleet with these). Each table shows a comparison of the two scenarios for the eleventh instance, each table contains the column **Routes (K=Auto-carriers)**, this shows the number of routes generated for the eleventh instance, the **Distance (KM)** column contains the cost in terms of kilometers of the routes and the **Time (Min)** column shows the cost in minutes of the same.

Table 6 concentrates the results of the routing of the instances using auto-carriers with a capacity 10 and 11 vehicles in the MDTW scenario. The use of capacity auto-carriers 11 obtains a 13% decrease in total distance and total time compared to capacity 10. In addition, 60 less auto-carriers were used to route the demand of the eleventh instance.

**Table 6.** Results of the main dealerships with different auto-carriers.

| | Auto-Carrier with $W = 10$ | | | Auto-Carrier with $W = 11$ | | |
|---|---|---|---|---|---|---|
| Instance | Routes (K) | Distance (Km) | Time (Min) | Routes (K) | Distance (Km) | Time (Min) |
| 1 | 4 | 19,263 | 12,660 | 4 | 19,263 | 12,660 |
| 2 | 7 | 18,329 | 12,481 | 7 | 18,338 | 12,493 |
| 3 | 18 | 71,366 | 48,256 | 14 | 62,571 | 41,809 |
| 4 | 24 | 81,429 | 55,950 | 21 | 71,231 | 48,997 |
| 5 | 69 | 148,588 | 102,617 | 59 | 130,604 | 87,928 |
| 6 | 137 | 323,696 | 213,737 | 118 | 270,872 | 179,242 |
| 7 | 195 | 417,909 | 277,747 | 170 | 364,385 | 241,743 |
| 8 | 254 | 550,555 | 367,032 | 220 | 471,456 | 312,449 |
| 9 | 318 | 681,336 | 458,476 | 278 | 585,245 | 389,968 |
| 10 | 366 | 794,553 | 525,284 | 323 | 696,878 | 459,669 |
| 11 | 478 | 1,037,633 | 686,562 | 418 | 900,562 | 597,259 |

Algorithm 2 was designed to work with a heterogeneous fleet of auto-carriers, the results of the tests in the two scenarios are shown in Table 7. In this table the results obtained from the tests are compared with the 11 instances in the two scenarios. Sometimes obtaining a smaller number of routes does not guarantee that it is the lowest total distance of the routes, e.g., in the row of instance 5 for the RDTW scenario, 95 routes are generated. In comparison with the 102 routes obtained in the MDTW scenario, but the total travel distance is 41,218 km smaller in this scenario (MDTW).

**Table 7.** Results of heterogeneous fleet (3,6,7,10 & 11).

| | Random Dealerships | | | Main Dealerships | | |
|---|---|---|---|---|---|---|
| Instance | Routes (K) | Distance (Km) | Time (Min) | Routes (K) | Distance (Km) | Time (Min) |
| 1 | 4 | 17,458 | 11,416 | 5 | 25,649 | 16,875 |
| 2 | 10 | 27,626 | 18,854 | 9 | 22,964 | 15,622 |
| 3 | 26 | 94,510 | 62,905 | 22 | 91,525 | 61,156 |
| 4 | 32 | 111,317 | 75,206 | 37 | 124,226 | 84,035 |
| 5 | 95 | 240,376 | 160,016 | 102 | 199,158 | 134,659 |
| 6 | 180 | 436,777 | 284,208 | 190 | 446,823 | 294,392 |
| 7 | 278 | 563,385 | 367,985 | 278 | 572,442 | 381,464 |
| 8 | 350 | 724,204 | 476,686 | 350 | 764,601 | 511,309 |
| 9 | 444 | 932,767 | 611,969 | 450 | 933,942 | 623,494 |
| 10 | 490 | 1,091,345 | 710,645 | 495 | 1,049,780 | 696,274 |
| 11 | 655 | 1,389,356 | 907,976 | 660 | 1,416,358 | 943,464 |

Regarding the assignment of vehicles, Algorithm 2 returned feasible loads as illustrated in Figure 9. Table 8, concentrates the data of 18 vehicles (VIN and height ($h_k$)) before entering the allocation phase.

**Table 8.** Vehicle data.

| VIN | Vehicle $h_k$ | VIN | Vehicle $h_k$ |
|-----|---------------|-----|---------------|
| 1 | 2.52 m | 10 | 1.47 m |
| 2 | 1.47 m | 11 | 1.47 m |
| 3 | 1.47 m | 12 | 1.47 m |
| 4 | 1.47 m | 13 | 1.47 m |
| 5 | 1.47 m | 14 | 1.47 m |
| 6 | 1.47 m | 15 | 1.47 m |
| 7 | 2.52 m | 16 | 1.47 m |
| 8 | 1.87 m | 17 | 1.47 m |
| 9 | 1.47 m | 18 | 1.47 m |

From Table 8, Figure 9a shows the output of Algorithm 2 using the auto-carriers of capacity 11 (odd), in which we can observe the allocation of two vans (VIN1, VIN7) using three spaces in both auto-carriers in their platforms, the rest of spaces are occupied by other vehicles. On the other hand, Figure 9b shows the output of the allocation of the vans in auto-carriers of capacity 6.

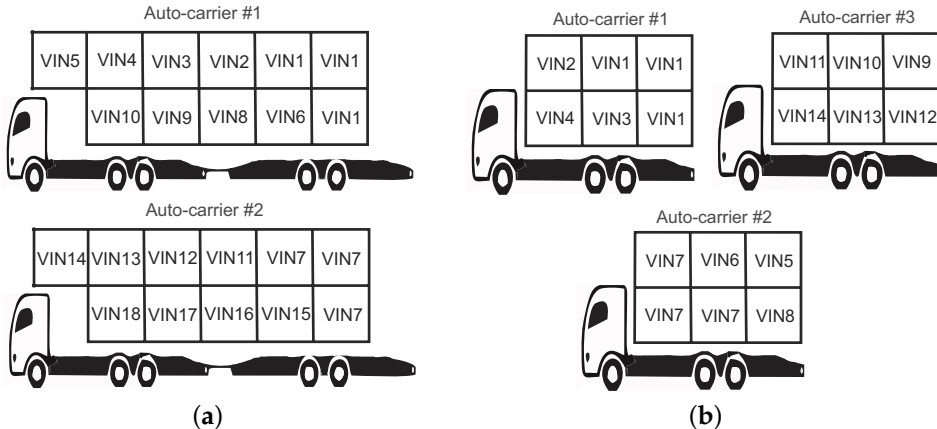

**Figure 9.** Output of vehicles allocation in auto-carriers: (**a**) Odd capacity (**b**) Pair capacity.

In Table 9 concentrates the results obtained to perform the routing of the demand in the MDTW scenario, using different capacities of auto-carriers and heterogeneous fleet, as it is highlighted in the capacity row of 11 vehicles, this shows the best results as regards distance (km) and time (min), as well as a lower number of auto-carriers (418) employed to carry out the routing of the real demand of the logistics company. A computer with Intel Core i5 7600K@3.8 GHz processor and 16 GB of RAM were used to perform the tests.

**Table 9.** Results of routing instance 11 with different auto-carriers.

| Auto-Carrier (Capacity) | Main Dealerships | | |
|-------------------------|------------------|----------------|------------|
| | Routes (K) | Distance (Km) | Time (Min) |
| 3 | 2043 | 4,146,291.8 | 2,720,157 |
| 6 | 936 | 2,018,547.9 | 1,332,574 |
| 7 | 688 | 1,488,546.3 | 980,416 |
| 10 | 478 | 1,037,633.2 | 686,562 |
| **11** | **418** | **900,562.8** | **597,259** |
| 3, 6, 7, 10 & 11 | 660 | 1,416,358.8 | 943,464 |

*6.2. Analysis of the Results*

The logistics company before the implementation of Algorithm 1 routed the auto-carriers empirically, i.e., the personnel in charge of this process did it without the assistance of some planning or optimization software, meaning the construction of inefficient routes [17]. Similarly, the loading of the vehicles in the auto-carrier was the responsibility of the operators, a process prone to damage during the unloading of the vehicles upon arrival at the dealership. Due to the absence of a LIFO policy that considers the route of the auto-carrier in the process of vehicle allocation.

The planning and routing of the auto-carriers of the logistics company were favorably impacted by the implementation of Algorithm 1, obtaining as output the generated routes (e.g., $d_0 - > d_{44} - > d_{35} - > d_0$) for the auto-carriers, the unloading vehicles in each dealership and auto-carrier schedules, as shown in Table 10. These results were possible to obtain thanks to the heuristic routing algorithm that considers the time (*CT* variable and time matrix). In addition, the proposed Algorithm 2 allowed to automate the process of allocation of vehicles in the auto-carrier, making it easier for operators to load the vehicles, examples of the output of this Algorithm 2 are shown in Figure 9. The allocation constraints can be adjusted to the requirements of different vehicles to be transported, but retaining the allocation logic.

**Table 10.** Example of schedule of the auto-carrier route.

| Dealership | Arrival | Departure | Unloaded Vehicles |
|:---:|:---:|:---:|:---:|
| d0 | –:– | 18:19 | 0 |
| d44 | 22:59 | 23:14 | 6 |
| d35 | 08:21 | 08:36 | 5 |
| d0 | 21:55 | –:– | 0 |

The efficiency of the proposed algorithm was demonstrated by being able to generate planning (routes, schedules, vehicle accommodations) in a reasonable time for more than 2000 routes, using auto-carriers of 3 vehicles of capacity. With the same performance, the results were obtained in a real case of the logistics company, using a heterogeneous fleet were generated 660 routes as shown in Table 9. As a consequence of the size of the demand, the routes constructed by Algorithm 1 contain from one dealership to four dealerships in their planning.

## 7. Conclusions and Future Research Works

The results of the experimental work with the proposed heuristic algorithm were satisfactory. These show the ability to route and obtain feasible loads for the auto-carriers with the demand of the logistics company. The allocation of vehicles using restrictions reduced the likelihood that the vehicles suffer some damage during the loading/unloading in the dealership, in addition to complying with the traffic guidelines that govern the auto-carriers in Mexico.

The implementation of the algorithm allowed obtaining the planning of the routes and the feasible loading of vehicles at a reasonable time, considering a demand of 4000 vehicles and 44 dealerships as a destination, which translates into thousands of kilometers diminished, i.e., a saving of fuel, money, and time for the logistics company, while polluting emissions are reduced. Impacting favorably in the decision-making regarding the planning and programming of the routing of the auto-carriers that has its service.

Future work is to develop tests with other auto-carrier capabilities, in addition to developing a metaheuristic algorithm, with the combination of the heuristic Algorithm 1 to obtain an initial solution and a PSO to improve the current solution. In addition to implementing a dynamic routing according to a

series of variables that can be presented in the current route of the auto-carrier, such as blocked routes or the transport of vehicles from one dealership to another.

**Author Contributions:** Conceptualization, O.A.P., P.M.Q.F. and R.E.P.L.; methodology and supervision, R.E.P.L. and P.M.Q.F.; data curation, software and writing–original draft preparation, M.A.J.P. and C.F.P.; investigation and writing–review and editing, M.A.J.P. and R.E.P.L.; validation and project administration, P.M.Q.F. and O.A.P.; all authors have read and approved the final manuscript.

**Funding:** This research received no external funding.

**Acknowledgments:** The first two authors acknowledge support from CONACyT through a scholarship to complete the Master in Computational Systems program at Tecnológico Nacional de México (TecNM). The authors appreciate the company for providing real data and the facilities granted for this research. In addition, the authors would like to thank the referees for their useful suggestions that helped to improve the quality of this paper.

**Conflicts of Interest:** The authors declare no conflict of interest.

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
