# Peer review of "A Heuristic Algorithm for the Routing and Scheduling Problem with Time Windows: A Case Study of the Automotive Industry in Mexico"

_algorithms, doi:10.3390/a12050111_

Round 1
Reviewer 1 Report
Authors study a real-world distribution problem arising in the vehicle production industry, particularly in a logistics company, in which cars and vans have to be loaded on auto-carriers and then delivered to dealerships. They propose heuristic solutions to address routing and scheduling problem in a brilliant manner. Generally, the idea is interesting, the results are convincing some minor suggestions are in order:
1. The language of the paper is not easy to read. It is better to take time and better present them.
2. Can we have computational complexity for the proposed heuristic algorithm?
3. Can we have tested on the scaled time window test case for your results?
4. I see some background are missing like "Using the gravitational emulation local search algorithm to solve the multi-objective flexible dynamic job shop scheduling problem in Small and Medium Enterprises" that can be of interest to add.
Author Response
Response to Reviewer 1 Comments
Authors study a real-world distribution problem arising in the vehicle production industry, particularly in a logistics company, in which cars and vans have to be loaded on auto-carriers and then delivered to dealerships. They propose heuristic solutions to address routing and scheduling problem in a brilliant manner. Generally, the idea is interesting, the results are convincing some minor suggestions are in order:
Point 1: The language of the paper is not easy to read. It is better to take time and better present them.
Response 1: Some paragraphs and sentences of the paper have been revised. Mainly in the results section. The word “scenery” was replaced by “scenario”.
Point 2: Can we have computational complexity for the proposed heuristic algorithm?
Response 2: The computational complexity of the proposed heuristic algorithm (O(n^2 log n^2)) was added to section 5, given that it is the complexity of the Solomon I1 insertion heuristic. Although the allocation algorithm has a complexity of O(n) because they are only nested IFs, since it is a subroutine added to the insertion heuristic, the complexity of the heuristic algorithm is summarized in (O(n^2 log n^2)).
Point 3: Can we have tested on the scaled time window test case for your results?
Response 3: The results obtained were tested with scaled time windows, Table 3 shows the format with which the time windows were configured and with the help of the eleven instances the two scenarios were tested.
Point 4: I see some background are missing like "Using the gravitational emulation local search algorithm to solve the multi-objective flexible dynamic job shop scheduling problem in Small and Medium Enterprises" that can be of interest to add.
Response 4: After a review of the literature on the subject presented, the suggested publication was added to section 2 with its corresponding citation.

Reviewer 2 Report
This work proposed a two-phase heuristic algorithm to solve the vehicle routing problem with time windows (VRPTW) in a logistics company in Mexico.
In the first phase the heuristic builds a route with an optimal insertion procedure, and in the second phase the determination of a feasible loading.
A real database of approximately 4000 vehicles, more than 600 auto-carriers and 44 different dealerships as a destination is used for experimentation.
Some more detailed comments are given below. I hope that if the authors will take them into account the paper will be improved.
1.How to find a set of proper initial parameters for the proposed algorithm? It seems to me that it is not an easy job and is really problem dependent to select the
parameters, since there are lots of parameters (mu, alpha, lumbda, level, parity, etc.) influencing the performance of the proposed method together.
2. Provide all algorithmic parameter setting of the proposed algorithm in Section 6.1.
3. Comparison of proposed methodology with the state-of-the-art techniques/its benefits. Please add a detailed discussion.
Author Response
Response to Reviewer 2 Comments
This work proposed a two-phase heuristic algorithm to solve the vehicle routing problem with time windows (VRPTW) in a logistics company in Mexico. In the first phase the heuristic builds a route with an optimal insertion procedure, and in the second phase the determination of a feasible loading. A real database of approximately 4000 vehicles, more than 600 auto-carriers and 44 different dealerships as a destination is used for experimentation.
Some more detailed comments are given below. I hope that if the authors will take them into account the paper will be improved.
Point 1: How to find a set of proper initial parameters for the proposed algorithm? It seems to me that it is not an easy job and is really problem dependent to select the parameters, since there are lots of parameters (mu, alpha, lambda, level, parity, etc.) influencing the performance of the proposed method together.
Response 1: In the paper "Algorithms for the Vehicle Routing and Scheduling Problems with Time Window Constraints" three heuristics are compared and evaluated, the insertion heuristic I1 is the one that obtains the best set of solutions for the VRPTW (the corresponding citation was added to section 5), the paper provides the appropriate initial values for the insertion heuristic I1, which were added in table 4 of section 6.1. The influence of these last two parameters on the algorithm results does not define the construction of the routes, only the total number of auto-carriers used to satisfy the demand.
Point 2: Provide all algorithmic parameter setting of the proposed algorithm in Section 6.1.
Response 2: Table 4 was added to section 6.1 with the parameters for the heuristic algorithm proposed.
Point 3: Comparison of proposed methodology with the state-of-the-art techniques/its benefits. Please add a detailed discussion.
Response 3: In section 5 the reference to research works that contemplate the comparison of the insertion heuristic I1 with other methods was added. In addition, the computational complexity of the heuristic algorithm was added to the section.

Reviewer 3 Report
The article describes the heuristic approach to the classical sheduling problem with time windows. The article is well-written and organized. However, I have a few comments:
1)It would be worth taking into account the following articles to improve the receipt of work
a)An efficient heuristic algorithm for the alternative-fuel station location problem, European Journal of Operational Research
b)Polar bear optimization algorithm: Meta-heuristic with fast population movement and dynamic birth and death mechanism, Symmetry
c)Bio-inspired voice evaluation mechanism, Applied Soft Computing
2)Authors use abbreviations without explaining them, eg LIFO.
3) It would be good to add a graphical illustration of the comparison with other methods.
4) There are no statistical tests at work.
5) Can the authors add a comparison of more heuristics from recent years?
Author Response
Response to Reviewer 3 Comments
The article describes the heuristic approach to the classical scheduling problem with time windows. The article is well-written and organized. However, I have a few comments:
Point 1: It would be worth taking into account the following articles to improve the receipt of work:
a) An efficient heuristic algorithm for the alternative-fuel station location problem, European Journal of Operational Research
b) Polar bear optimization algorithm: Meta-heuristic with fast population movement and dynamic birth and death mechanism, Symmetry
c)Bio-inspired voice evaluation mechanism, Applied Soft Computing
Response 1: After a review of the literature on the subject presented, the suggested publications were added to section 2 with their corresponding citation.
Point 2: Authors use abbreviations without explaining them, e.g. LIFO.
Response 2: In section 5.1 we add the explanation of the abbreviation LIFO (Last In, First Out) and describe an example of the policy.
Point 3: It would be good to add a graphical illustration of the comparison with other methods.
Response 3: In section 5 the reference to research works that contemplate the comparison of the insertion heuristic I1 with other methods was added. In addition, the computational complexity of the heuristic algorithm was added to the section.
Point 4: There are no statistical tests at work.
Response 4: Statistical work was done to compare the cost of the routes according to the capacity of the auto-carrier, the results are presented in Tables 5 and 8. However, no statistical tests were performed to evaluate the insertion heuristic I1 because it was not the objective of the investigation.
Point 5: Can the authors add a comparison of more heuristics from recent years?
Response 5: In section 5 the reference to the work "Vehicle routing with time windows: An overview of exact, heuristic and metaheuristic methods" was added, in which the insertion heuristic I1 compares with other methods.

Round 2
Reviewer 2 Report
The authors have carefully addressed the previous comments of the reviewer and significantly improved the manuscript.
Reviewer 3 Report
Authors made all corrections. It can be acccepted in current form.